# Serum Leucine-Rich α2 Glycoprotein Could Be a Useful Biomarker to Differentiate Patients with Normal Colonic Mucosa from Those with Inflammatory Bowel Disease or Other Forms of Colitis

**DOI:** 10.3390/jcm13102957

**Published:** 2024-05-17

**Authors:** Ichitaro Horiuchi, Kaori Horiuchi, Akira Horiuchi, Takeji Umemura

**Affiliations:** 1Department of Gastroenterology, Shinshu University Hospital, Matsumoto 390-8621, Japan; ichitaro617@yahoo.co.jp (I.H.); tumemura@shinshu-u.ac.jp (T.U.); 2Digestive Disease Center, Showa Inan General Hospital, Komagane 399-4117, Japan; koraki648@yahoo.co.jp

**Keywords:** leucine-rich α2 glycoprotein, LRG, biomarker, IBD, colitis

## Abstract

(1) **Background:** Serum leucine-rich α2 glycoprotein (LRG) has been reported as a useful biomarker for monitoring disease activity in patients with inflammatory bowel disease (IBD). We investigated whether serum LRG can differentiate patients with normal colonic mucosa from those with IBD or other forms of colitis. (2) **Methods:** Patients with diarrhea, abdominal pain, or bloody stools were consecutively enrolled at their initial visit to our hospital. Serum LRG and C-reactive protein were measured, and a colonoscopy and histology were performed. (3) **Results:** We enrolled 317 patients (181 men, 136 women; median age: 51 years). Based on the endoscopic and histological criteria, 260 patients were diagnosed with ulcerative colitis (n = 134), Crohn’s disease (n = 10), infectious colitis (n = 43), diverticular colitis (n = 17), or nonspecific colitis (n = 56). The remaining 57 patients were diagnosed with normal colonic mucosa including histology. The latter group’s median LRG value (9.5 µg/mL, range: 5.8–13.5) was significantly lower than that of the other 260 patients (13.6 µg/mL, range: 6.8–62.7, *p* < 0.0001). The optimal LRG cut-off value of <10.4 µg/mL was derived from the receiver operating characteristic (ROC) curve, showing a 91% sensitivity and 77% specificity for identifying patients with normal colonic mucosa. (4) **Conclusions:** serum LRG values < 10.4 µg/mL could be a useful biomarker for predicting patients with normal colonic mucosa.

## 1. Introduction

Many individuals with diarrhea, abdominal pain, and/or bloody stool visit outpatient clinics. They may develop inflammatory bowel disease (IBD), as well as other forms of colitis that can macroscopically resemble IBD (i.e., infectious colitis, ischemic colitis, pseudomembranous colitis, colitis related to diverticular disease, drug colitis, allergic colitis, and microscopic colitis), each of which are characterized by their clinical, endoscopic, and histological features [1,2]. Although an endoscopic evaluation and a mucosal biopsy including a histological examination and the culture of colonic specimens are essential to diagnose IBD and non-IBD colitis, it is sometimes challenging to perform a colonoscopy examination with mucosal biopsy in routine clinical practice. In addition, a positive diagnosis of irritable bowel syndrome can be made based on the evaluation of gastrointestinal symptoms (especially alarm signs), normal colonic mucosa [3,4], and the use of the symptom-based Rome IV criteria [5]. Very recent study clearly showed that the Calprotectin immunoassay has the best value in discriminating between IBD and diarrhea-predominant irritable bowel syndrome [6]. An effective serum biomarker other than the routine measurement of patients’ C-reactive protein (CRP) is thus desired as a diagnostic tool to predict the presence or absence of IBD or other forms of colitis in outpatient clinics at individuals’ initial visits.

Leucine-rich α2 glycoprotein (LRG) has been isolated from human serum and characterized [7]; it is a 50 kDa glycoprotein that contains repetitive sequences with a leucine-rich motif [8]. Serada et al. used a proteomic approach and identified LRG as a novel biomarker for both rheumatoid arthritis and IBD [9]. LRG is an acute-phase protein, and its excessive expression has been associated with various inflammatory states in the development of respiratory, hematological, endocrine, malignancy, eye, cardiovascular, rheumatic immune, and infectious diseases [10]. LRG is induced by multiple inflammatory cytokines, such as tumor necrosis factor-alpha (TNF-α) and interleukin (IL)-22, IL-1β, and IL-6 [11].

LRG has been described by several research groups as a biomarker for detecting IBD and for monitoring disease activity in individuals with IBD [11,12,13,14]. We have demonstrated that a serum LRG cut-off value of ≥10.8 µg/mL could be used as a biomarker for predicting the presence of active ulcerative colitis (other than proctitis) in patients [15]. In addition, a recently published study has clearly shown that LRG in serum is a promising new biomarker that can distinguish acute appendicitis from cases without appendicitis in patients with nonspecific abdominal pain [16]. Another study even showed that the biomarker can be successfully obtained from saliva, eliminating the need for venipuncture, which could be important for the pediatric population [17]. A commercial LRG assay is now available for routine clinical use in Japan, where LRG can be used as a serum biomarker in patients with IBD. We conducted the present study as the first attempt to determine whether serum LRG values can be used to differentiate patients with normal colonic mucosa (including histology) from patients with IBD or non-IBD colitis at their initial visit.

## 2. Materials and Methods

### 2.1. Study Design and Setting

This prospective, controlled study was performed at the Digestive Disease Center, Showa Inan General Hospital, Japan. The study was approved by the local institutional Ethics Committee of our hospital (No. 2019-8) on 23 September 2019. All patients provided their written informed consent for participation when their enrollment in the study was scheduled. The study is registered in the Clinical Trials.gov registry under identifier NCT 04535882, and part of its results has been reported [15]. All authors had access to the study data and reviewed and approved the final manuscript.

### 2.2. Patients

Between October 2020 and July 2023, 2996 patients who presented to our department at Showa Inan General Hospital with diarrhea, abdominal pain, or bloody stool and no history of IBD were evaluated at their initial visits. A total of 335 patients suspected of IBD or other forms of colitis were consecutively enrolled. The exclusion criteria were the presence of a long-term illness, colorectal cancer, or ischemic colitis. Ten patients who had ischemic colitis, five who had colorectal cancer, and three who were taking lansoprazole were excluded. Finally, this study included 317 patients who had undergone a colonoscopy examination (Figure 1).

### 2.3. Measurement of the Partial Mayo Score, Serum CRP, and LRG Levels

In accord with the design of our prior study [14], the partial Mayo score was evaluated in all patients at their initial visit. For the evaluation of clinical symptoms, each patient’s clinical activity was evaluated using the partial Mayo score used for ulcerative colitis (active state was defined as a score of ≥2 points). Blood sampling and the colonoscopy examination including histology were performed within a few days after the patients’ first visit to our department. The CRP levels were measured using the latex immunoturbidimetric method (Shino-Test Corporation, Tokyo, Japan). Serum LRG levels were measured by latex turbidimetric immunoassay (Sekisui Medical, Tokyo, Japan) [11,12].

### 2.4. Assessment of Colonoscopy with Histology and Culture, and Abdominal CT Scan

Irrespective of the endoscopic findings, such as erythema, erosion, or ulceration, a colonic biopsy specimen was collected at four or more sites in the mucosa of each patient’s ascending, transverse, and descending colon and rectum. In addition, a culture of colonic biopsy specimens from the site of erythema, erosion, or ulceration was performed. In the patients with active ulcerative colitis (i.e., score ≥ 1 point), the endoscopic activity was graded using the Mayo endoscopic subscore [18]. The patients with ulcerative colitis were classified based on the extent of their disease involvement as having proctitis, left-sided colitis, or pancolitis as described by the Montreal classification [19]. A histological analysis was determined using Matts classification [20].

Apart from IBD, other forms of colitis, such as infectious colitis, colitis related to diverticular disease, ischemic colitis, eosinophilic colitis, pseudomembranous colitis, and microscopic colitis, were diagnosed based on their clinical, endoscopic and histological features with a culture of colonic biopsy specimens [1,2]. The patients with minor endoscopic findings such as erythema, erosion, and aphtha in whom normal flora were observed in the culture of their colonic biopsy specimens were eventually diagnosed with nonspecific colitis. An abdominal computed tomography (CT) scan was performed on demand.

### 2.5. Outcome Measures

The primary outcome measure of this study was to determine the LRG cut-off value for predicting patients with normal colonic mucosa. The secondary outcome measures were to compare the area under the receiver operating characteristic curve (AUC) values of LRG with those of the partial Mayo score and CRP.

### 2.6. Sample Size Calculation 

The calculation of the necessary sample size was based on the study’s primary outcome measure. Assuming a statistical power of 80% and a significance level of 0.05, AUC for the null hypothesis of 0.50 and 0.69 for the alternative hypothesis, and a ratio of positive to negative cases of normal colonic mucosa of 2, a minimum of 82 subjects (at least 41 per group) must be drawn from the consecutive sampling of patients presenting to our department with possible colitis.

### 2.7. Statistical Analyses

Data are presented as the median (range). The χ^2^-test (with Yates’ correction for continuity where appropriate) was used for the comparisons of categorical data. Fisher’s exact test was used when the numbers were small. For parametric data, a Student’s *t*-test was used when two means were compared. Nonparametric data were analyzed by the Mann–Whitney U test when two medians were compared. We used receiver operating characteristic (ROC) curves and the AUC to examine the relationships between the presence/absence of each form of colitis including IBD and the partial Mayo score, the CRP value, and the LRG value. The optimal threshold levels of the partial Mayo score, CRP, and LRG were each calculated using the Youden index. A probability (*p*)-value < 0.05 was regarded as significant. The software GraphPad Prism ver. 9.3.1 (Boston, MA, USA) was used for the statistical analysis.

## 3. Results

### 3.1. Patients’ Clinical Characteristics

We enrolled 317 patients (181 men, 153 women; median age: 51 years); 260 patients were diagnosed with ulcerative colitis (n = 134), Crohn’s disease (colonic or ileocolonic) (n = 10), infectious colitis (n = 43), diverticular colitis (n = 17), or nonspecific colitis (n = 56) based on the results of their endoscopic and histological examinations, including colonic mucosal cultures. The remaining 57 patients (18%) were eventually diagnosed as having normal colonic mucosa including histology.

The clinical characteristics of the enrolled patients are summarized in Table 1. There was a significant difference in gender between the patients with normal colonic mucosa (normal) and patients with Crohn’s disease (*p* = 0.04) but not significantly different between normal and other groups. The ages of the patients with normal colonic mucosa were significantly younger than those of the patients with ulcerative colitis, infectious colitis or diverticular colitis (*p* < 0.01), but they were not significantly different from those of the patients with Crohn’s disease or nonspecific colitis. There were significant differences in symptom durations between normal and patients with Crohn’s disease, infectious colitis, or diverticular colitis (*p* < 0.01) but not significantly different between normal and other groups. The median CRP and LRG values in the normal-mucosa group were significantly lower than those of the IBD and other IBD-colitis groups, respectively, as follows: CRP, 0.03 mg/dL vs. 0.17–0.45 mg/dL, *p* < 0.01; LRG, 9.5 µg/mL vs. 13.6–22.1 µg/mL, *p* < 0.01. The median LRG level of normal control volunteers at our digestive disease center (n = 20) was 10.2 µg/mL.

### 3.2. Histological Findings including Culture Results

Table 2 summarizes the histological findings of Matts classification in the enrolled patients. All 57 patients with normal colonic mucosa showed normal (grade 1) histology. Forty-three patients with infectious colitis had grade 2 (21%) or 3 (79%) histology, and the cultures of colonic biopsy specimens revealed enterohemorrhagic *Escherichia coli* (n = 22), *Campylobacter* (n = 13), *Salmonella* (n = 4), and *Yersinia* (n = 4). Fifty-two of the fifty-six patients with nonspecific colitis (93%) showed mild inflammation (grade 2), but normal flora were also observed in the cultures of colonic biopsy specimens. No patients who fulfilled the definition of microscopic colitis were identified in this study.

### 3.3. Comparison of Patients with Normal Colonic Mucosa and Those with IBD or Another Form of Colitis

Figure 2 depicts the results of the comparison of the partial Mayo score, CRP values, and LRG values between the patients with normal colonic mucosa and those with IBD or another form of colitis (including ulcerative colitis, Crohn’s disease, infectious colitis, diverticular colitis, and nonspecific colitis). The median partial Mayo score, CRP value, and LRG value of the normal colonic mucosa group were each significantly lower compared to those of the other enrolled patients (*p* < 0.0001).

### 3.4. ROC Curves for the Partial Mayo Score, CRP, and LRG for Predicting Normal Colonic Mucosa

We evaluated the diagnostic ability of LRG for predicting patients with normal colonic mucosa including histology in a comparison with the partial Mayo score and the CRP value by determining the area under the curve (AUC) of the ROC curves (Figure 3). The AUC for the LRG values was significantly higher than the AUC of both the partial Mayo score (*p* < 0.0001) and the CRP values (*p* < 0.0001). The optimal threshold value of LRG was <10.4 µg/mL from the respective ROC curve, showing 91% sensitivity, 77% specificity, a positive predictive value of 47%, and a negative predictive value of 98%. These sensitivity, specificity, positive predictive, and negative predictive values were higher than those of both the partial Mayo score and the CRP value (Table 3), and these data demonstrated that the LRG value is more sensitive than that of CRP for predicting patients with normal colonic mucosa including histology, irrespective of some symptoms.

## 4. Discussion

The symptoms diarrhea, abdominal pain, and bloody stools are each frequent reasons for visits to an outpatient clinic or emergency department. These may be symptoms of an underlying organic disease or a functional disorder without an anatomic or physiologic alteration. Evaluations of patients with diarrhea, abdominal pain, and/or bloody stools are a challenge for primary physicians, and the selection of patients for second-level radiological examinations or endoscopic procedures is sometimes difficult. A diagnostic marker that could be used to distinguish between organic and functional disorders has, thus, been required for further examinations in routine clinical practice or in the context of an emergency setting.

The results of the present study demonstrated that using the serum LRG cut-off value of <10.4 µg/mL could be a useful biomarker to predict patients with normal colonic mucosa (including histology) at their initial visit to a clinic or emergency department without performing a colonoscopy examination. Based on our findings, physicians could easily identify individuals who need to undergo a colonoscopy exam as soon as possible, even if they have reported persistent diarrhea, abdominal pain, or bloody stools. This study appears to provide the first report indicating the effectiveness of a serum LRG level that can be used to exclude patients with inflammation of the colon and rectum (including IBD) at their initial visit.

It can sometimes be difficult to distinguish irritable bowel syndrome with diarrhea (which is a representative disease of patients without colonic inflammation with histology) from IBD, and symptoms alone cannot always be used to accurately distinguish the two diseases [21,22]. Fifty-six patients with minor endoscopic findings such as erythema, erosion, and aphtha in whom normal flora were observed in the culture of their colonic biopsy specimens were eventually diagnosed with nonspecific colitis. Although the present patients had diarrhea, abdominal pain, or bloody stools at their initial visits, 57 patients with normal colonic mucosa including histology (18% of the total patient series) were identified in this study. Finally, some of these 57 patients had been diagnosed with irritable bowel syndrome with diarrhea. 

The measurement of a patient’s CRP value is the serological test most commonly used to exclude IBD in patients with irritable bowel syndrome [23]. A comprehensive meta-analysis that evaluated serological markers in 2145 subjects identified as having IBD or irritable bowel syndrome plus healthy controls revealed that a CRP value of ≤0.5 mg/dL predicted a ≤1% probability of IBD with good accuracy [24]. In the present study, the CRP cut-off value of <0.05 mg/dL for predicting patients with normal colonic mucosa such as irritable bowel syndrome roughly corresponded to this previous result. Two fecal-derived markers of intestinal inflammation (i.e., fecal lactoferrin and fecal calprotectin) are both diagnostically useful and perhaps superior to serologic tests (e.g., the erythrocyte sedimentation rate and CRP) based on their diagnostic accuracy in discriminating IBD from irritable bowel syndrome with diarrhea [4,25]. However, stool sampling is troublesome for patients, and methods to evaluate fecal lactoferrin and fecal calprotectin are often not available in clinical practice. We thus did not examine these two fecal-derived markers of intestinal inflammation in the present study. The advantage of LRG may be beyond those of CRP, fecal lactoferrin, and fecal calprotectin to exclude patients with inflammation of the colon and rectum (including IBD) in routine clinical practice.

LRG has been described as a useful biomarker for detecting Crohn’s disease at the optimal cut-off value of ≥14 µg/mL [14,26,27,28]. Our present analyses demonstrated that a serum LRG cut-off value of ≥10.8 µg/mL could be used as a biomarker for predicting active ulcerative colitis (other than proctitis) in a portion of the enrolled patients [15]. However, the negative predictive value for Crohn’s disease was highest (0.71) with an 89% sensitivity at an LRG cut-off value of 9 µg/mL [29], and the mean LRG level in a Crohn’s disease subgroup of patients with small-bowel mucosal healing was 10.7 [30] or 10.0 µg/mL [31]. Another study showed that among IBD patients including patients with ulcerative colitis, the mean LRG levels at endoscopic remission were 11.2 µg/mL at 12 weeks and 11.7 µg/mL at 52 weeks after the introduction of adalimumab treatment [13]. In ulcerative colitis patients at complete endoscopic remission, the optimal LRG cut-off value was 10.0 µg/mL [32]. Our present finding that the optimal LRG threshold for predicting patients with normal colonic mucosa, which is <10.4 µg/mL, is compatible with the above-reported LRG level of IBD patients with mucosal healing or endoscopic remission, i.e., 10.0 µg/mL. However, we could not compare the LRG values found in patients with IBD using Nanopia^®^ LRG kit, (Sekisui Medical, Tokyo, Japan) [13,14,15] and those values published in patients with other inflammatory diseases, such as acute appendicitis [16]. Because commercially available kits used for the measurement of serum LRG in these studies were different.

Within this study, patients with normal colonic mucosa were significantly younger in comparison to those with abnormal mucosa, except for patients with Crohn’s disease or nonspecific colitis (Table 1). It is possible that serum LRG levels were higher in healthy older adults, which may influence the result of the comparison. However, we could not compare the levels of LRG during between the younger patients with normal colonic mucosa and the younger patients with IBD or another form of colitis in this study. In the future, the additional study may lead to more reliable results by reducing the potential confounding effect of age on LRG levels.

This study has some limitations. It was conducted at a single hospital in Japan, and in addition to the small number of normal controls (n = 20), the number of patients with normal colonic mucosa was small (n = 57). In addition, the exact diagnoses of 56 patients who were diagnosed with nonspecific colitis were unknown. As the partial Mayo score, the Mayo endoscopic subscore, and Matts classification are scores evaluating activity of ulcerative colitis, these scores may not be preferable in evaluating other diseases than ulcerative colitis. It was also not possible to eliminate the influence of the patients’ backgrounds or patient selection bias. Larger, randomized multicenter trials with larger patient populations are necessary to test our findings.

## 5. Conclusions

Using a serum LRG cut-off value of <10.4 µg/mL could be a useful biomarker for predicting patients with normal colonic mucosa including histology at their initial visit.

## Figures and Tables

**Figure 1 jcm-13-02957-f001:**
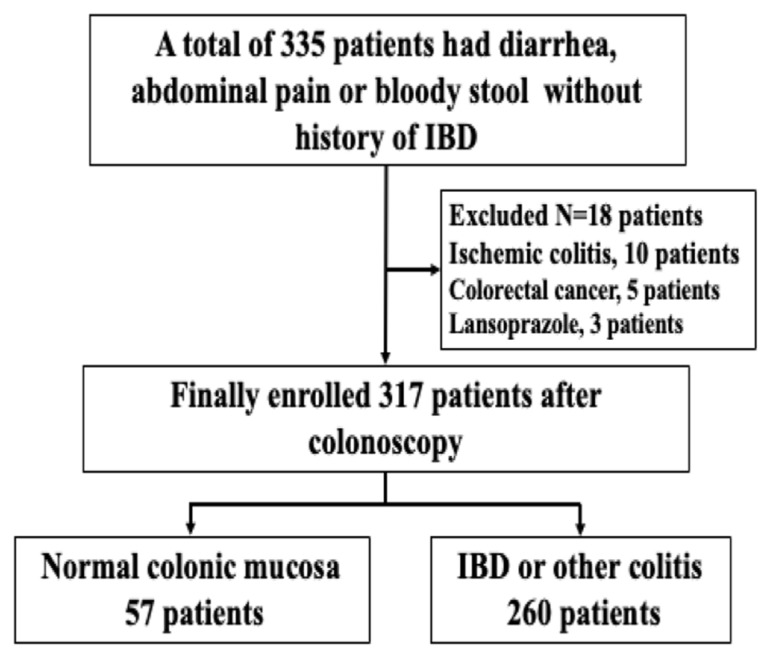
Study flow diagram of enrolled patients. IBD: inflammatory bowel disease.

**Figure 2 jcm-13-02957-f002:**
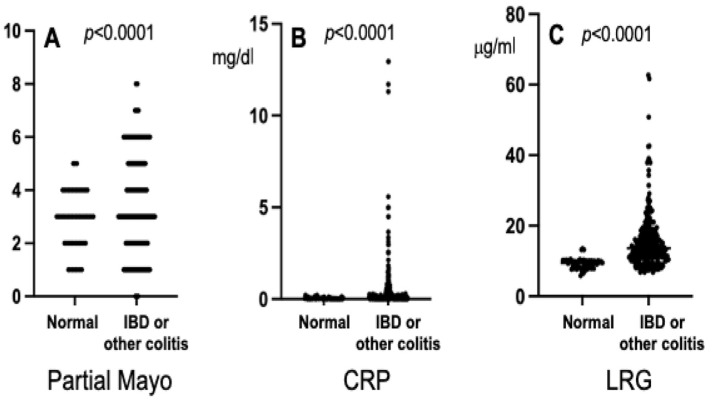
Comparison of each value of the partial Mayo score (**A**), C-reactive protein (CRP) (**B**), and leucine-rich α2 glycoprotein (LRG) (**C**) between the group of patients with normal colonic mucosa (Normal) and the group of patients with inflammatory bowel disease (IBD) or another form or colitis (IBD or other colitis).

**Figure 3 jcm-13-02957-f003:**
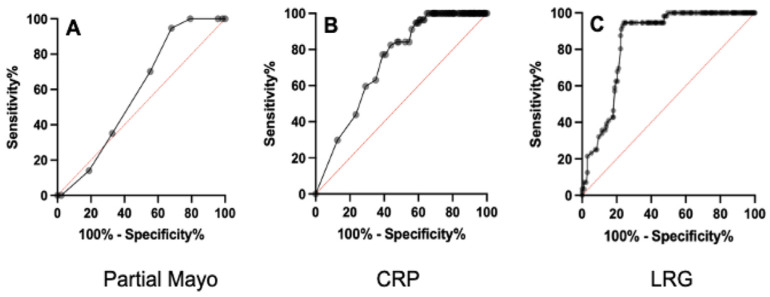
Receiver operating characteristic (ROC) curves for the partial Mayo score (**A**), CRP (**B**), and LRG (**C**) for predicting patients with normal colonic mucosa including histology.

**Table 1 jcm-13-02957-t001:** Comparison of clinical characteristics between the patients with normal colonic mucosa and the patients with inflammatory bowel disease (IBD) or another form of colitis.

	Normal n = 57	Ulcerative Colitis n = 134	Crohn’s Disease n = 10	Infectious Colitis n = 43	Diverticular Colitis n = 17	Nonspecific Colitis n = 56
Males/females, n (%)	32 (56)/25 (44)	74 (55)/60 (45)	7 (70)/3 (30)	25 (58)/18 (42)	11 (65)/6(35)	28 (50)/28 (50)
Age, mean, years (range)	35 (18–49)	52 (18–79)	32 (21–65)	43 (23–55)	65 (45–77)	34 (19–55)
Symptom duration, days	12 (8–40)	14 (7–45)	65 (34–88)	3 (1–5)	12 (6–22)	4 (2–10)
WBC, 10^9^/L	5.7 (4.3–10.7)	7.7 (4.4–14.7)	4.7 (3.4–12.1)	7.7 (5.4–14.5)	5.3 (3.7–13.1)	5.7 (3.7–15.4)
Hemoglobin, g/dL	14.2 (12.3–14.5)	13.2 (10.3–14.5)	12.2 (10.3–13.4)	14.1 (12.5–14.4)	13.2 (11.3–16.4)	13.8 (10.3–14.4)
Platelets, 10^9^/dL	43.5 (23–39.5)	32.5 (23–77.5)	42.5 (33–67.6)	46.5 (36–57.8)	32.5 (27–57.6)	43.5 (35–65.6)
Albumin, g/dL	3.6 (3.5–4.4)	3.2 (2.9–4.5)	3.4 (3.1–4.3)	3.3 (3.1–4.3)	3.1 (2.9–4.1)	3.5 (3.2–4.4)
Partial Mayo score	3 (1–5)	4 (1–8)	3 (1–5)	5 (2–7)	3 (1–4)	3 (1–4)
CRP, mg/dL	0.03 (0.01–0.21)	0.25 (0.01–13)	0.25 (0.01–13)	0.45 (0.2–11)	0.35 (0.01–10.5)	0.17 (0.01–8.5)
LRG, µg/mL	9.5 (5.8–13.5)	13.6 (6.8–62.7)	17.2 (13.8–23.5)	22.1 (15.8–45.3)	16.5 (13.8–42.7)	18.6 (12.6–51.5)

Normal: patients with normal colonic mucosa; WBCs: white blood cells. The data are median (range) unless otherwise noted.

**Table 2 jcm-13-02957-t002:** Comparison of histological findings between the patients with normal colonic mucosa and those with inflammatory bowel disease (IBD) or another form of colitis.

Matts Classification *, Grade	Normal n = 57	Ulcerative Colitis n = 134	Crohn’s Disease n = 10	Infectious Colitis n = 43	Diverticular Colitis n = 17	Nonspecific Colitis n = 56
1	57 (100%)	0	5 (50%)	0	3 (18%)	4 (7%)
2	0	0	5 (50%)	9 (21%)	13 (76%)	52 (93%)
3	0	23 (17%)	0	34 (79%)	1 (6%)	0
4	0	44 (33%)	0	0	0	0
5	0	67 (50%)	0	0	0	0

* Based on the patient’s histopathological findings. Normal: patients with normal colonic mucosa.

**Table 3 jcm-13-02957-t003:** The area under the curve (AUC), sensitivity, and specificity of the partial Mayo score, CRP, and LRG for predicting patients with normal colonic mucosa.

Variable	AUC (95%CI)	Cut-Off Value	Sensitivity (%, 95%CI)	Specificity (%, 95%CI)	PPV (%, 95%CI)	NPV (%, 95%CI)
Partial Mayo	0.59 (0.52–0.65)	<3.5	70 (57–81)	45 (39–51)	21 (16–27)	87 (76–93)
CRP	0.73 (0.67–0.79)	<0.055 mg/dL	77 (65–86)	61 (55–67)	22 (17–88)	89 (82–95)
LRG	0.84 (0.79–0.88)	<10.4 µg /mL	91 (81–96)	77 (72–82)	47 (36–55)	98 (87–99)

AUC: area under the curve; CI: confidence interval; CRP: C-reactive protein; LRG: leucine-rich α2 glycoprotein; NPV: negative predictive value; partial Mayo: partial Mayo score; PPV: positive predictive value.

## Data Availability

The data underlying this article will be shared upon reasonable request to the corresponding author.

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
