# Peer review of "Serum Leucine-Rich α2 Glycoprotein Could Be a Useful Biomarker to Differentiate Patients with Normal Colonic Mucosa from Those with Inflammatory Bowel Disease or Other Forms of Colitis"

_jcm, 2024, doi:10.3390/jcm13102957_

Round 1
Reviewer 1 Report
Comments and Suggestions for Authors
The present study by Ichitaro Horiuchi and colleagues looked at patients with or without inflammatory bowel disease (IBD) or other forms of colitis. The study aims to address whether serum LRG cutoff value (<10.4 ug/ml) can be used to identify a patient with a normal-appearing gut despite having colitis symptoms. This method could potentially help predict patients with normal colonic mucosa at their initial visit, a concept that has not been extensively investigated in previous research. Researchers compared the levels of LRG in the blood with disease activity scores (partial Mayo score) and CRP (another marker of inflammation) and proved that the LRG cutoff is a good way to distinguish patients with normal colonic mucosa. The rationale for the study is clearly explained in the introduction, highlighting the urgency for better methods to identify patients who may not need a colonoscopy exam due to low levels of serum LRG. The results affirm the potential of commercial LRG assay to predict normal colonic mucosal with good accuracy, a finding that could significantly improve patient management.
The research is promising. However, there are several areas for improvement:
Major Comments:
-
It has been shown that LRG levels tend to increase with age. Within this study, individuals with normal colonic mucosa were notably younger in comparison to those with abnormal mucosa. It is possible that LRG levels were higher in healthy older adults, which may influence the result of the comparison. The authors should consider the potential benefits of creating more comparable groups. For instance, they could compare the levels of LRG in the blood in each group where the age range is below 50 years. This could lead to more robust and reliable results by reducing the potential confounding effect of age on LRG levels. Additionally, this would bolster the author's assertion that LRG stands out as the optimal biomarker for excluding patients with an inflamed colon.
Minor Comments:
-
In Table 1, there seems to be a discrepancy in the unit of LRG concentration. It is listed as mg/mL, but based on the context and standard practice, the correct unit should be ug/mL. This clarification is important for the accuracy of the data presented.
-
Statistic-wise (Line125), the Mann-Whitney U Test is a nonparametric statistical test. Hence, the author should specify what data is parametric data and what is non-parametric data. This will help the readers understand the appropriateness of the statistical method used in the study.
In Materials and Methods, there are similarities between the paper published by the same authors in 2022 and the current text. These similarities are significant and cannot be ignored. The authors may consider rephrasing them.
Author Response
Responses to Reviewer 1:
Major Comments:
- It has been shown that LRG levels tend to increase with age. Within this study, individuals with normal colonic mucosa were notably younger in comparison to those with abnormal mucosa. It is possible that LRG levels were higher in healthy older adults, which may influence the result of the comparison. The authors should consider the potential benefits of creating more comparable groups. For instance, they could compare the levels of LRG in the blood in each group where the age range is below 50 years. This could lead to more robust and reliable results by reducing the potential confounding effect of age on LRG levels. Additionally, this would bolster the author's assertion that LRG stands out as the optimal biomarker for excluding patients with an inflamed colon.
Response: Thank you for your review of our manuscript. Your comments and suggestions have helped us improve the text. The following text was added to the Discussion section in response to your comments.
Page 11: Within this study, patients with normal colonic mucosa were significantly younger in comparison to those with abnormal mucosa except for patients with Crohn’s disease or nonspecific colitis (Table 1). It is possible that serum LRG levels were higher in healthy older adults, which may influence the result of the comparison. However, we could not compare the levels of LRG during between the younger patients with normal colonic mucosa and the younger patients with IBD or another form of colitis in this study. In the future, the additional study may lead to more reliable results by reducing the potential confounding effect of age on LRG levels.
Minor Comments:
- In Table 1, there seems to be a discrepancy in the unit of LRG concentration. It is listed as mg/mL, but based on the context and standard practice, the correct unit should be ug/mL. This clarification is important for the accuracy of the data presented.
Response: In Table 1 and Table 3, mg/mL was changed to ug/mL in the unit of LRG concentration.
- Statistic-wise (Line125), the Mann-Whitney U Test is a nonparametric statistical test. Hence, the author should specify what data is parametric data and what is non-parametric data. This will help the readers understand the appropriateness of the statistical method used in the study.
The sentence “Both parametric data and nonparametric data were analyzed by the Mann-Whitney U-test when two medians were compared.” was changed to the following.
Page 4: For parametric data, a Student’s t-test was used when two means were compared. Non-parametric data were analyzed by the Mann–Whitney U-test when two medians were compared.
Comments on the Quality of English Language
In Materials and Methods, there are similarities between the paper published by the same authors in 2022 and the current text. These similarities are significant and cannot be ignored. The authors may consider rephrasing them.
According to your comments, we rephrased them especially in the revised manuscript.

Reviewer 2 Report
Comments and Suggestions for Authors
The authors investigated whether the leucine-rich α2-glycoprotein (LRG) in serum can distinguish patients with normal colonic mucosa from those with inflammatory bowel disease (IBD) or other forms of colitis. They concluded that serum LRG levels <10.4 µg/mL may be a useful biomarker for predicting patients with normal colonic mucosa.
The manuscript is well organized and of interest, methodology and study design are appropriate and reproducible. The presentation of the results is well done and adequately discussed. The main methodological shortcoming is the lack of sample size calculation. The literature is up to date. However, several important points should be revised/addressed before a positive decision can be made.
1. Abstract – Please expand the abbreviation "ROC". In addition, I would suggest adding the term "LRG" under the keywords.
2. Introduction – The authors state that C-reactive protein is an effective biomarker for predicting the presence or absence of IBD. I agree, but the authors should add a few lines regarding diagnostic value of Calprotectin. Very recent study clearly showed that the Calprotectin immunoassay has the best value in discriminating between IBD and IBS-D (10.5455/medarh.2024.78.105-111).
3. Introduction – The authors state that LRG is an acute-phase protein and its overexpression has been associated with various inflammatory conditions in the development of respiratory, hematologic, endocrine, malignant, ocular, cardiovascular, rheumatic, immunologic, and infectious diseases. In my opinion, it is of great importance to emphasize that LRG can be elevated in other gastrointestinal inflammatory diseases besides IBD, such as acute appendicitis. A recently published study has clearly shown that LRG in serum is a promising new biomarker that can distinguish acute appendicitis from cases without appendicitis in patients with non-specific abdominal pain (10.3390/jcm12072455). Another study (diagnostic accuracy of leucine-rich α-2-glycoprotein 1 as a non-invasive salivary biomarker in pediatric appendicitis) even showed that the biomarker can be successfully obtained from saliva, eliminating the need for venipuncture, which could be important for the pediatric population.
4. Outcomes of the study are not clear. Please add a new paragraph on methodology and describe the primary and secondary outcomes of the study.
5. It is unclear which statistical test was used to test the normality of the data distribution. Please clarify.
6. The main methodological objection to this study is the lack of sample size calculation. As this is a prospective study, sample size calculation is mandatory. Please comment and clarify this point! If this was not done, the authors should clearly state why and add this to the study limitations. If a sample size calculation was performed, please include this as a separate paragraph in the methodology.
7. The authors have used paragraph "2.2." twice in the methodology. Please revise (2.2. Patients and 2.2. Measurement of the partial Mayo score, serum CRP and LRG levels).
8. The authors state that CRP was routinely measured. Please add further details on which method and kit was used.
9. The key demographic data of all groups should be compared to check whether the groups were symmetrical in terms of baseline demographic parameters. Please include a statistical comparison of the groups for baseline data.
10. Discussion – The authors should compare the LRG values found in patients with IBD (their own or published studies) with values published in patients with other inflammatory diseases, such as acute appendicitis, to check whether these values are significantly different.
Comments on the Quality of English LanguageMinor editing required.
Author Response
Reviewer 2
- Abstract – Please expand the abbreviation "ROC". In addition, I would suggest adding the term "LRG" under the keywords.
Response: Thank you for your review of our manuscript. Your comments and suggestions have helped us improve the text. I expanded the abbreviation “receiver operating characteristic (ROC)” and “LRG” was added to keywords.
- Introduction – The authors state that C-reactive protein is an effective biomarker for predicting the presence or absence of IBD. I agree, but the authors should add a few lines regarding diagnostic value of Calprotectin. Very recent study clearly showed that the Calprotectin immunoassay has the best value in discriminating between IBD and IBS-D (10.5455/medarh.2024.78.105-111).
Response: The following sentence was added to the introduction section and a paper was added to the references.
Page 1: Very recent study clearly showed that the Calprotectin immunoassay has the best value in discriminating between IBD and diarrhea-predominant irritable bowel syndrome [6].
- Huong BT, Hien NM, Dung NT, et al. Role of Calprotectin, IL-6, and CRP in distinguishing between inflammatory bowel disease and diarrhea predominant irritable bowel syndrome. Med Arch. 2024; 78: 105-111.
- Introduction – The authors state that LRG is an acute-phase protein and its overexpression has been associated with various inflammatory conditions in the development of respiratory, hematologic, endocrine, malignant, ocular, cardiovascular, rheumatic, immunologic, and infectious diseases. In my opinion, it is of great importance to emphasize that LRG can be elevated in other gastrointestinal inflammatory diseases besides IBD, such as acute appendicitis. A recently published study has clearly shown that LRG in serum is a promising new biomarker that can distinguish acute appendicitis from cases without appendicitis in patients with non-specific abdominal pain (10.3390/jcm12072455). Another study (diagnostic accuracy of leucine-rich α-2-glycoprotein 1 as a non-invasive salivary biomarker in pediatric appendicitis) even showed that the biomarker can be successfully obtained from saliva, eliminating the need for venipuncture, which could be important for the pediatric population.
Response: The following sentences were added to the introduction section and two papers were added to the references.
Page 2: In addition, a recently published study has clearly shown that LRG in serum is a promising new biomarker that can distinguish acute appendicitis from cases without appendicitis in patients with non-specific abdominal pain [16]. Another study (diagnostic accuracy of leucine-rich α-2-glycoprotein 1 as a non-invasive salivary biomarker in pediatric appendicitis) even showed that the biomarker can be successfully obtained from saliva, eliminating the need for venipuncture, which could be important for the pediatric population [17].
- Tintor G,Jukić M,Šupe-Domić D, et al. Diagnostic utility of serum leucine-rich α-2-glycoprotein 1 for acute appendicitis in children. J Clin Med 2023; 12: 2455.
17.Tintor G, Jukić M, Šupe-Domić D, et al. Diagnostic accuracy of leucine-rich α-2-glycoprotein 1 as a non-invasive salivary biomarker in pediatric appendicitis. Int J Mol Sci. 2023; 24:6043.
- Outcomes of the study are not clear. Please add a new paragraph on methodology and describe the primary and secondary outcomes of the study.
Response: The following sentences were added to the method section.
Page 3: 2.5. Outcome measures  The primary outcome measure of this study was to determine the LRG cutoff value for predicting for predicting patients with normal colonic mucosa. The secondary outcome measures were to compare the area under the receiver operating characteristic curve (AUC) values of LRG with those of the partial Mayo score and CRP.
- It is unclear which statistical test was used to test the normality of the data distribution. Please clarify.
Response: The sentence “Both parametric data and nonparametric data were analyzed by the Mann-Whitney U-test when two medians were compared.” was changed to the following.
Page 4: For parametric data, a Student’s t-test was used when two means were compared. Non-parametric data were analyzed by the Mann–Whitney U-test when two medians were compared.
- The main methodological objection to this study is the lack of sample size calculation. As this is a prospective study, sample size calculation is mandatory. Please comment and clarify this point! If this was not done, the authors should clearly state why and add this to the study limitations. If a sample size calculation was performed, please include this as a separate paragraph in the methodology.
Response: The following sentences were added as a separate paragraph in the method section.
Page 4: 2.6. Sample Size Calculation
The calculation of the necessary sample size was based on the study's primary outcome measure. Assuming a statistical power of 80% and a significance level of 0.05, the area under the curve (AUC) for the null hypothesis of 0.50 and 0.69 for the alternative hypothesis, and a ratio of positive to negative cases of normal colonic mucosa of 2, a minimum of 82 subjects (at least 41 per group) must be drawn from the consecutive sampling of patients presenting to our department with possible colitis.
- The authors have used paragraph "2.2." twice in the methodology. Please revise (2.2. Patients and 2.2. Measurement of the partial Mayo score, serum CRP and LRG levels).
Response: Paragraph used 2.2 twice was corrected in the revised manuscript.
- The authors state that CRP was routinely measured. Please add further details on which method and kit was used.
Response: The following sentence was added to the method section.
Page 3: The CRP levels were measured using the latex immunoturbidimetric method (Shino-Test Corporation, Tokyo, Japan).
- The key demographic data of all groups should be compared to check whether the groups were symmetrical in terms of baseline demographic parameters. Please include a statistical comparison of the groups for baseline data.
Response: The following sentences were added to the results section.
Page 4: There was a significant difference in gender between the patients with normal colonic mucosa (normal) and patients with Crohn’s disease (p=0.04), but not significantly different between normal and other groups. The ages of the patients with normal colonic mucosa were significantly younger than those of the patients with ulcerative colitis, infectious colitis or diverticular colitis (p<0.01), but they were not significantly different from those of the patients with Crohn’s disease or nonspecific colitis. There were significant differences in symptom durations between normal and patients with Crohn’s disease, infectious colitis or diverticular colitis (p<0.01), but not significantly different between normal and other groups.
- Discussion – The authors should compare the LRG values found in patients with IBD (their own or published studies) with values published in patients with other inflammatory diseases, such as acute appendicitis, to check whether these values are significantly different.
Response: The following sentences were added to the discussion section.
Page 8: However, we could not compare the LRG values found in patients with IBD (their own or published studies)[13,14,15] and those values published in patients with other inflammatory diseases, such as acute appendicitis [16]. Because commercially available kits used for the measurement of serum LRG in these studies were different.

Round 2
Reviewer 2 Report
Comments and Suggestions for Authors
The authors significantly improved the manuscript after revision. However, two minor points should be revised / addressed prior to any favorable decision is made:
The authors added a statement: Another study (diagnostic accuracy of leucine-rich α-2-glycoprotein 1 as a non-invasive salivary biomarker in pediatric appendicitis) even showed that the biomarker can be successfully obtained from saliva, eliminating the need for venipuncture, which could be important for the pediatric population [17]. There is no need to present a title of a study in brackets. Reference number should be enough. Please revise.
A following sentence should be revised for clarity: However, we could not compare the LRG values found in patients with IBD (their own or published studies) [13,14,15] and those values published in patients with other inflammatory diseases, such as acute appendicitis [16]. – ‘’their own or published studies’’ – this statement is unclear and should be rewritten for clarity.
Comments on the Quality of English Language-
Author Response
May 15, 2024
Editor
JCM Editorial Office
Manuscript ID: jcm-3009087 “Serum leucine-rich α2 glycoprotein could be a useful biomarker to differentiate patients with normal colonic mucosa from those with inflammatory bowel disease or other forms of colitis”
Dear Editors:
Thank you very much for a very nice review. According to your comments we have revised the manuscript with point-by-point responses to the comments and concerns noted.
We hope you will be satisfied with the revised manuscript.
Yours sincerely,
Akira Horiuchi, M.D.
Chief
Digestive Disease Center
Showa Inan General Hospital
Reviewer’s comments (Round 2):
The authors added a statement: Another study (diagnostic accuracy of leucine-rich α-2-glycoprotein 1 as a non-invasive salivary biomarker in pediatric appendicitis) even showed that the biomarker can be successfully obtained from saliva, eliminating the need for venipuncture, which could be important for the pediatric population [17]. There is no need to present a title of a study in brackets. Reference number should be enough. Please revise.
Response: Thank you for your review of our manuscript. According to your comment, this sentence was changed to the following.
Page 1: Another study even showed that the biomarker can be successfully obtained from saliva, eliminating the need for venipuncture, which could be important for the pediatric population [17].
A following sentence should be revised for clarity: However, we could not compare the LRG values found in patients with IBD (their own or published studies) [13,14,15] and those values published in patients with other inflammatory diseases, such as acute appendicitis [16]. – ‘’their own or published studies’’ – this statement is unclear and should be rewritten for clarity.
Response: According to your comment, this sentence was changed to the following.
Page 9: However, we could not compare the LRG values found in patients with IBD using Nanopia® LRG kit, (Sekisui Medical, Tokyo, Japan) [13,14,15] and those values published in patients with other inflammatory diseases, such as acute appendicitis [16].
